# Foundation Model Ensemble for Out-of-Distribution Generalization: Predicting Lymph Node Metastasis in Early Gastric Cancer Using Whole-Slide Imaging

**Woojin Chung**[1]              GOGLXYCH97@HUFS.AC.KR

[1] *Deparment of Biomedical Engineering, Hankuk University of Foreign Studies, Yongin-si 17035, Gyeonggi-do, Korea.*

**Yujun Park**[2]              ISUTAR_STAR@NAVER.COM

[2] *Department of Pathology, CHA Bundang Medical Center, CHA University, Seongnam-si 13496, Gyeonggi-do, Korea.*

**Yoonho Nam**[1]              YOONHONAM@HUFS.AC.KR

**Editors:** Accepted for publication at MIDL 2025

## Abstract

Recent advances in deep learning have improved the practicality of automated analysis for whole-slide imaging. However, challenges remain in image analysis due to variations in imaging equipment, tissue preparation, staining protocols, and other variables. These variations hinder the generalizability of trained models to external datasets. Recently, foundation models trained on large-scale pathology datasets have been introduced by various research groups, demonstrating the potential to address this issue. Since each foundation model was trained on datasets collected from different sources under varying settings, the learned representations reflect different characteristics to some extent. These differences suggest that leveraging the information of multiple models could improve generalization and robustness compared to using a single model. In this study, we investigate foundation model ensembles for predicting lymph node metastasis in early gastric cancer across three different datasets. By comparing ensemble models with individual ones, we demonstrate that ensembling multiple foundation models improves performance in whole-slide imaging for both in-distribution and out-of-distribution data.

**Keywords:** Whole-Slide Imaging (WSI), Foundation Model, Foundation Model Ensemble, Lymph Node Metastasis Prediction, Early Gastric Cancer

## 1. Introduction

The development of computational pathology has made whole-slide imaging (WSI) an essential tool in pathology diagnosis and research (Aeffner et al., 2019; Kumar et al., 2020). WSI enables high-resolution scanning of pathology slides, converting them into digital images for efficient computational processing and analysis. Although these advancements have greatly enhanced computational pathology, variability in pathological images—arising from differences in imaging equipment, tissue processing, staining protocols, and other variables—limits model generalizability across datasets (Aeffner et al., 2019).

Foundation models, trained on diverse datasets through self-supervised learning, have the potential to address such limitations by learning generalized representations. These

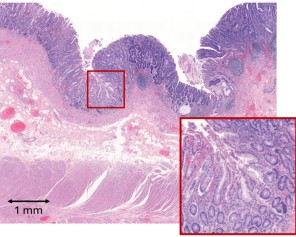
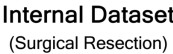
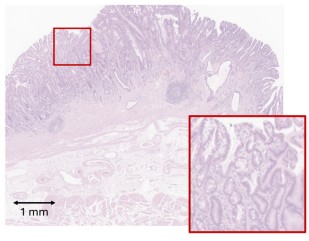
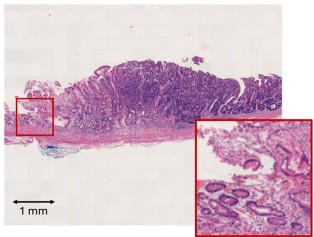

**Internal Dataset**
(Surgical Resection)

**External Dataset**
(Surgical Resection)

**ESD Dataset**
(Endoscopic Resection)

Figure 1: Although all these images are WSIs from EGC patients, they show differences in imaging characteristics, including color tone and intensity, depending on the acquisition methods or resection methods.

representations capture complex patterns in training data, enabling robust performance on external datasets. Recently, foundation models trained on large-scale pathology datasets have been introduced by various research groups, demonstrating their adaptability to downstream tasks (Ciga et al., 2022; Wang et al., 2022; Chen et al., 2022; Filiot et al., 2023; Vorontsov et al., 2023; Hua et al., 2024; Alfasly et al., 2024; Chen et al., 2024; Lu et al., 2024; Nechaev et al., 2024; Xu et al., 2024; Yang et al., 2025). Each foundation model, trained on datasets collected from different sources and under varying settings, develops its own representations that reflect these differences, leading to varied benchmark performances (Wölflein et al., 2023). This diversity suggests that each foundation model may contribute complementary information to downstream tasks.

The complementary information captured by different foundation models could enhance generalization and robustness when combined, particularly on external datasets. Previous studies showed that transforming the features of a foundation model using information from another produced more generalized results compared to relying on a single model (Chung et al., 2024). This insight highlights the potential of foundation model ensembles for overcoming limitations in model generalization due to dataset variations.

Ensemble methods, which combine models to improve overall performance, have been widely utilized in machine learning to address the limitations of single models by reducing variance (Mohammed and Kora, 2023). Beyond variance reduction, ensembling foundation models can integrate their unique information derived from pre-trained data, further enhancing generalization. However, the efficacy of foundation model ensembles in the context of WSI remains underexplored, with limited research conducted in this area.

In this study, we investigate the effectiveness of foundation model ensembles for predicting lymph node metastasis (LNM) in early gastric cancer (EGC) WSIs, including performance on out-of-distribution (OOD) datasets. We evaluated model generalizability using two test datasets with different distributions: one varying in data acquisition and another in resection methods. A detailed description is provided in Section 2.1. In our case, these OOD datasets are contextually relevant but exhibit distinct characteristics in color tone and intensity (Figure 1). Following the definition by Farquhar and Gal, they can be categorized as related distributions, a subset of OOD. By ensembling multiple foundation models, we aim to explore their contribution to model performance and generalization in WSI analysis.

## 2. Method

LNM in gastric cancer is closely associated with characteristics of tumor areas (Maruyama et al., 1989). Therefore, to predict LNM in EGC, we first trained patch-level classification networks to extract tumor regions from WSIs. Within these regions, we trained patch-level classification networks to predict the LNM using slide-level labels. These predictions were then aggregated into a slide-level representation, and ensemble methods were applied using the top-performing models on the validation dataset and were compared across datasets.

### 2.1. Datasets

This study used three datasets—internal, external, and endoscopic submucosal dissection (ESD)—categorized based on the acquiring institution and treatment type. The internal dataset was selected from patients at our institution who underwent curative surgical resection with lymph node dissection for EGC. The external dataset comprised surgical cases for EGC collected from different institutions and scanned with a different scanner. The ESD dataset included endoscopic resection cases with subsequent lymph node dissection from multiple institutions. We split the internal dataset into training and validation sets, using the training set for model training and the validation set for ensemble validation. The external and ESD datasets were used as test sets. Table 1 provides a summary of the datasets. (Note: LN+ and LN- indicate counts of WSIs with and without LNM, respectively.)

Table 1: Summary of the datasets.

| Dataset | Split | LN+ | LN- | Institution | Treatment |
|---------|-------|-----|-----|-------------|-----------|
| Internal Dataset | Train | 100 | 100 | Internal Institution | Curative surgical resection with lymph node dissection |
| | Valid | 30 | 73 | | |
| External Dataset | Test | 30 | 71 | External Institution | |
| ESD Dataset | Test | 23 | 96 | Internal + External | Endoscopic resection cases with subsequent lymph node dissection |

### 2.2. Pre-trained Models

In this study, we selected 13 pre-trained models for ensemble learning—one ImageNet pre-trained model and 12 foundation models in computational pathology. The computational pathology foundation models included Ciga et al. (Ciga et al., 2022), CTransPath (Wang et al., 2022), HIPT (Chen et al., 2022), Phikon (Farquhar and Gal, 2022), Virchow (Vorontsov et al., 2023), PathoDuet (Hua et al., 2024), PathDINO (Alfasly et al., 2024), UNI (Chen et al., 2024), CONCH (Lu et al., 2024), Hibou (Nechaev et al., 2024), Prov-GigaPath (Xu et al., 2024), BEPH (Yang et al., 2025) with detailed descriptions provided in Table 2.

### 2.3. Tumor Region Extraction

A pathologist annotated the tumor areas in 80 WSIs, 40 cases with and 40 cases without LNM. Using these annotations, we fine-tuned three separate foundation models by Ciga et al. for patch-level classification of cancer regions at three magnifications ($20\times$, $10\times$,

Table 2: Details of the Pre-trained Models Used

| Name | Model Architecture | Trained Method | Trained Dataset | Feature Dim |
|------|-------------------|----------------|-----------------|-------------|
| ImageNet | ResNet34 | Supervised Learning | 1.2M natural images | 768 |
| Ciga et al. | ResNet18 | SimCLR | 206K patches + 25K WSIs from multiple sources | 512 |
| CTransPath | CNN + Swin ViT | SRCL | 15M patches from 30K WSIs (TCGA and PAIP) | 768 |
| HIPT | Three hierarchical ViT | DINO | 10,678 WSIs, 104M 256x256 images, 408K 4096×4096 images | 384 |
| Phikon | ViT-B | iBOT | 43M patches and 6K WSIs from PanCancer40M (including TCGA-COAD, PanCancer4M) | 768 |
| Virchow | ViT-H | DINOv2 | 1.5M WSIs (MSKCC) | 1280 |
| Pathoduet | ViT-B | MoCov3 + SimSiam + InfoNCE Loss | 11K WSIs (TCGA), 2771 pair WSIs (HyReCo) and 3896 pair WSIs(BCI) | 768 |
| PathDINO | Lightweight ViT | DINO + HistoRotate | 6M patches from 11K WSIs (TCGA) | 384 |
| UNI | ViT-L | DINOv2 | 100M patches from 100K WSIs (Mass-100K) | 1024 |
| CONCH | Transformer-based | CoCa | 1.17M image-caption pairs | 512 |
| Hibou | ViT-B and ViT-L | DINOv2 | 1.2B patches (L), 512M patches (B) | 768 |
| Prov-GigaPath | LongNet | DINOv2 + MAE | 1.38B tiles from 171K WSIs (Prov-Path) | 1536 |
| BEPH | ViT-B | BEiT | ImageNet-1k and 11M patches from 11,760 pathology images (TCGA) | 768 |

and 5×). We averaged the probability maps generated by the three models and applied a threshold ($> 0.5$) to extract the tumor regions from the WSIs. A pathologist reviewed the extracted regions across the entire dataset and confirmed their appropriateness. These validated regions were then used for LNM prediction.

## 2.4. Data Processing for LNM Prediction

Since each foundation model requires a different input image size based on its trained settings, we first tiled the extracted tumor regions into 512×512 pixels with an overlap ratio of 0.5 between adjacent tiles. Then, the tiles were randomly cropped into patches matching the input size required by each foundation model (224×224 pixels for scratch-trained networks). Each foundation model also required a distinct normalization method; we therefore applied the corresponding normalization parameters. Aside from these differences, the same data augmentation strategies were consistently applied across all training processes.

## 2.5. Training Single Models for LNM Prediction

For each model described in Section 2.2, we trained a patch-level classifier using the tumor regions extracted in Section 2.3. Each classifier consisted of three non-linear layers followed by a sigmoid activation and was trained under identical settings. For baseline comparisons, ResNet34 (He et al., 2016) and ViT-Base (Dosovitskiy et al., 2020) were trained from scratch. All LNM prediction networks were trained and evaluated at 10× magnification. Each trained model generated an LNM risk probability map for each WSI. Slide-level LNM predictions were obtained by averaging the top 100 patches with the highest risk scores from the probability map, which were selected experimentally.

## 2.6. Foundation Model Ensembles for LNM Prediction

Based on the single models trained in Section 2.5, we selected the top-performing models using their performance on the internal validation set. Specifically, we constructed ensembles using the top 3 and top 5 models, respectively. For each of these subsets, we applied three different ensemble strategies for LNM prediction: (1) Soft voting for slide-level classification, (2) Averaging probability maps to aggregate patch-level predictions, and (3) Feature concatenation, where extracted features from multiple models were combined and fed into a classification network.

In addition, we compared single-model ensembles with multi-model ensembles. For the highest-performing foundation model based on internal validation performance, we implemented soft voting and averaging probability maps.

## 2.7. Evaluating Model Calibration, Uncertainty and Consistency

To further understand the advantages of foundation model ensembles, we quantified model calibration, uncertainty and consistency. Calibration was measured by the brier score (BS) (Brier, 1950), and uncertainty was measured using the widely adopted negative log-likelihood (NLL) (Lakshminarayanan et al., 2017), defined as:

$$\text{BS} = \frac{1}{N} \sum_{i=1}^{N} (\hat{y}_i - y_i)^2, \quad \text{NLL} = -\frac{1}{N} \sum_{i=1}^{N} \left[ y_i \log(\hat{y}_i) + (1 - y_i) \log(1 - \hat{y}_i) \right].$$

BS and NLL values were compared using predictive scores at both the patch-level within cancer regions and the slide-level of WSIs.

Model consistency was evaluated by analyzing the stability of predictions across patches with similar histopathological features in 28 true positive LNM cases from the internal dataset. For these cases, a pathologist carefully annotated the top 20 patches per case based on three primary criteria: (1) Tumor Differentiation and Main Types, (2) Inflammatory Response, and (3) Stromal and Tissue Features, resulting in a total of 560 annotated patches. To evaluate the consistency of predictions for each patch category, we calculated the standard deviation of the prediction scores.

## 3. Result

### 3.1. Evaluation of Single Models for LNM Prediction

The area under the curve (AUC) was used as the evaluation metric. Table 3 presents the AUC scores for LNM prediction for individual models on the Internal, External, and ESD datasets. Even when trained on the same data using same conditions(e.g. learning parameters, loss functions, and data augmentations), the results varied across the models. Figure 2 shows representative examples of LNM predictions for different models.

On the Internal Dataset, most foundation models demonstrated comparable or superior performance to the scratch-trained models. The highest-performing model in our downstream task was Ciga. et al., achieving an AUC score of 0.867.

On the external dataset, however, only BEPH (AUC 0.796) as feature extractors outperformed scratch-trained ViT-base (AUC 0.786), which was trained from scratch. This

Table 3: AUC Scores for Different Models Across Datasets

| Model Name | Internal Dataset | External Dataset | ESD Dataset |
|---|---|---|---|
| Scratch ResNet34 | 0.839 | 0.748 | 0.538 |
| Scratch ViT-base | 0.827 | 0.786 | 0.471 |
| ImageNet (ResNet34) | 0.794 | 0.760 | 0.641 |
| Ciga et al. | **0.867** | 0.742 | 0.663 |
| CTransPath | 0.859 | 0.778 | **0.702** |
| HIPT | 0.802 | 0.701 | 0.600 |
| Phikon | 0.841 | 0.763 | 0.628 |
| Virchow | 0.858 | 0.754 | 0.655 |
| Pathoduet | 0.740 | 0.734 | 0.510 |
| PathDINO | 0.847 | 0.700 | 0.534 |
| UNI | 0.853 | 0.718 | 0.631 |
| CONCH | 0.816 | 0.700 | 0.688 |
| Hibou | 0.843 | 0.779 | 0.597 |
| Prov-GigaPath | 0.835 | 0.646 | 0.621 |
| BEPH | 0.827 | **0.796** | 0.604 |

\* The best model in each dataset is highlighted in bold.

result may suggest that freezing the parameters of foundation models as feature extractors limits their ability to adapt their internal representations.

On the ESD dataset, scratch-trained models performed poorly (AUC 0.538 and 0.471), while the foundation models exhibited a range of AUC scores (0.510–0.702). The highest-performing models were CTransPath (0.702), CONCH (0.688), and Ciga et al. (0.663).

### 3.2. Evaluation of Foundation Model Ensembles for LNM Prediction

We selected the top-performing individual foundation models on the internal validation dataset for ensembling. Specifically, the top-3 models were Ciga et al., CTransPath, and Virchow, while the top-5 included UNI and PathDINO in addition. For each groups, ensemble methods, including soft voting, averaging probability maps, and feature concatenation, were used, and the AUC results for each method are presented in Table 4.

Table 4: Result AUC of Foundation Model Ensemble

|  | Method | Internal | External | ESD |
|---|---|---|---|---|
| | Scratch ResNet34 | 0.839 | 0.748 | 0.538 |
| | Scratch ResNet34 (Soft Voting) | 0.809 | 0.747 | 0.559 |
| | Scratch ResNet34 (Averaging Probability Maps) | 0.811 | 0.747 | 0.556 |
| | Ciga. et al. | 0.867 | 0.742 | 0.663 |
| Single Model | Ciga. et al. (Soft Voting) | 0.863 | 0.741 | 0.639 |
| | Ciga. et al. (Averaging Probability Maps) | 0.864 | 0.743 | 0.638 |
| | BEPH | 0.827 | 0.796 | 0.604 |
| | CTransPath | 0.859 | 0.778 | 0.702 |
| Top-3 Ensemble | Soft Voting | **0.886** | 0.761 | 0.702 |
| | Averaging Probability Maps | 0.879 | **0.806** | **0.714** |
| | Feature Concatenation | 0.848 | 0.718 | 0.678 |
| Top-5 Ensemble | Soft Voting | 0.883 | 0.746 | 0.682 |
| | Average Probability Map | 0.876 | 0.763 | 0.680 |
| | Feature Concatenation | 0.827 | 0.673 | 0.628 |

\* The best model in each dataset is highlighted in bold.

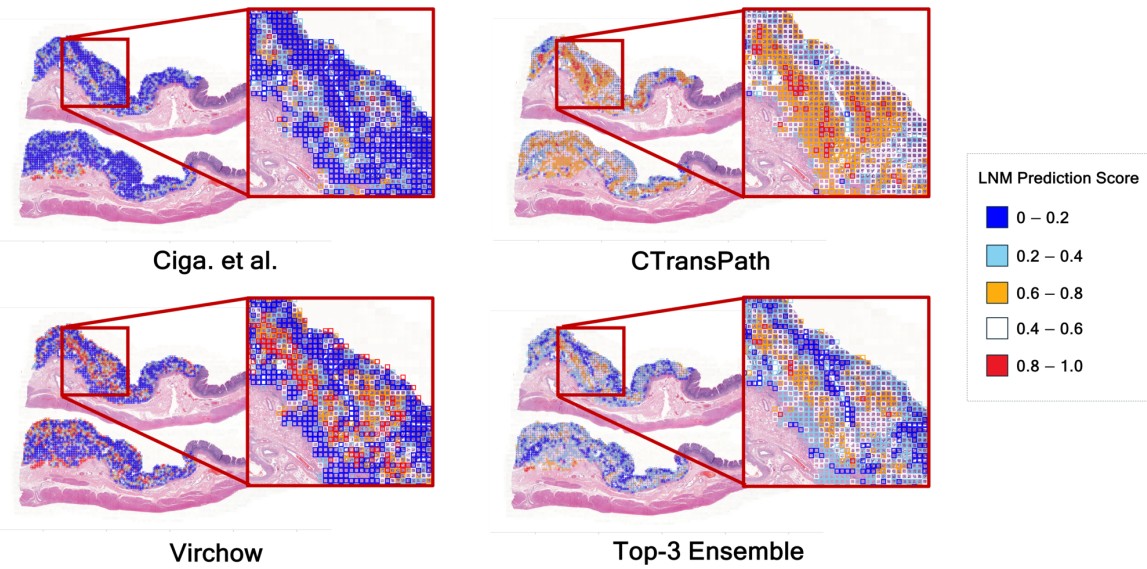

Figure 2: This figure shows examples of prediction heatmaps generated by individual foundation models and the top-3 ensemble using averaging probability maps.

As shown in Table 4, both the top-3 and top-5 ensembles demonstrated improved performance on the internal dataset when soft voting and averaging probability maps were applied. Notably, top-3 averaging the probability maps of the consistently outperformed both the individual foundation models and the scratch-trained models across all datasets. The top-5 ensemble showed lower performance compared to the top-3 ensemble across datasets in our setting. Moreover, single-model ensembles did not show significant improvements in performance.

### 3.3. Top-3 Averaging Probability Maps Model Calibration

We evaluated the BS at both the patch and slide levels for the top-3 ensemble using averaging probability maps, which demonstrated improved performance on the two OOD datasets. We compared the ensemble model's performance to that of its individual constituent models—Ciga et al., CTransPath, and Virchow. While the individual models exhibited variations in BS across different datasets, the ensemble method overall maintained lower values, indicating well-calibrated results.

At the patch-level, the ensemble achieved a BS of 0.234 in the internal dataset, compared to 0.235, 0.298, and 0.281 from individual models. In the external dataset, it achieved 0.224, while individual models achieved 0.229, 0.374, and 0.244. In the ESD dataset, the ensemble achieved 0.210, compared to 0.244, 0.206, and 0.285.

At the slide-level, the ensemble achieved a BS of 0.244 in the internal dataset, compared to 0.330, 0.208, and 0.456 from individual models. In the external dataset, it achieved 0.184, while individual models recorded 0.219, 0.221, and 0.224. In the ESD dataset, the ensemble achieved 0.448, compared to 0.465, 0.510, and 0.720.

### 3.4. Top-3 Averaging Probability Maps Model Uncertainty

To further evaluate the effectiveness of the foundation model ensemble, we evaluated uncertainty using NLL at both the patch and slide levels. The top-3 ensemble using averaging probability maps consistently exhibited lower uncertainty across all datasets compared to individual models (Ciga. et al., CTransPath, and Virchow).

At the patch-level, the ensemble achieved an NLL of 0.692 in the internal dataset, lower than 1.075, 0.692, and 1.003 from individual models. In the external dataset, the ensemble had 0.641, compared to 1.402, 0.657, and 0.686. Similarly, in the ESD dataset, the ensemble had 0.633, while individual models showed 0.727, 0.703, and 1.117.

At the slide-level, the ensemble method continued to demonstrate reduced uncertainty. In the internal dataset, the ensemble achieved 0.680, while individual models recorded 1.537, 0.867, and 0.602. In the external dataset, the ensemble had 0.552, compared to 0.638, 0.630, and 0.692. In the ESD dataset, where overall uncertainty was higher, the ensemble still showed the lowest NLL (1.166) compared to 3.169, 1.191, and 1.369.

### 3.5. Top-3 Averaging Probability Maps Model Consistency

To assess the consistency of the probability map ensemble, we analyzed the standard deviation of prediction scores across commonly observed patch categories. The two most frequent categories were "Moderately differentiated" (19.64%) and "Poorly differentiated" (11.43%). Examples of these patch categories can be found in Figure 3.

For the top-3 foundation models that achieved the best performance on the internal dataset (CTransPath, Virchow, and Ciga et al.), the standard deviation of risk probability scores for the "Moderately differentiated" category was 0.069, 0.045, and 0.082, respectively. When using the ensemble method, which combined the probability maps of these top-3 models, the standard deviation decreased to 0.042, indicating improved consistency.

For the "Poorly differentiated" category, the standard deviation for the top-3 models was 0.066, 0.053, and 0.027, respectively. Using the ensemble method, the standard deviation was reduced to 0.033. In this category, a decreased value was observed for two of the models, further supporting the effectiveness of ensembling in stabilizing predictions.

### 4. Discussion and Conclusion

In this paper, we evaluated the performance of individual foundation models and the effectiveness of foundation model ensembles for LNM prediction in EGC in three different EGC datasets. Among the ensemble methods, averaging the probability maps of the top-3 high-performing models in our task consistently improved performance across all datasets. This suggests that using ensemble methods to integrate information from multiple models in the context of WSI analysis is beneficial for improving overall performance. However, in averaging probability maps, using five models instead of three resulted in reduced performance, underscoring the importance of selecting foundation models suited to the ensemble approach and the downstream task.

The method of concatenating features and retraining has shown inferior performance compared to averaging probability maps. This may be attributed to the use of a simple

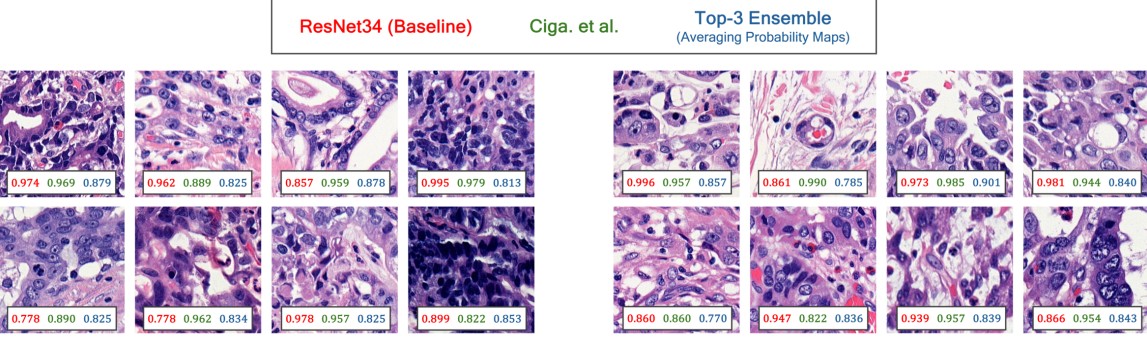

Figure 3: This figure shows the two most frequently observed categories among the top patches for LNM prediction: (left) Moderately differentiated with no significant inflammation, and (right) Poorly differentiated with no significant inflammation. The predicted scores from different models(scratch-trained ResNet34, Ciga et al., and averaging probability maps) are displayed for each patch.

three-layer non-linear classifier, which likely struggled to effectively capture relevant information within the increased dimensionality of the feature space. The higher dimensionality from concatenation may have led to noise or redundancy. These findings highlight the need for more efficient methods to integrate features across foundation models.

The effectiveness of the foundation model ensemble was indirectly evaluated through model calibration, uncertainty and consistency. The ensemble method reduced calibration and uncertainty, and stabilized predictions within the patch category. These results suggest that combining the information of individual foundation models could enhance generalization and robustness, including performance on OOD datasets.

There are several limitations in our study. First, foundation models were used solely as feature extractors, which may have constrained their ability to generalize, as seen in the test results on the external dataset. Due to computational constraints in our experimental setting, fine-tuning large models in their entirety was challenging. Future studies should explore fine-tuned foundation model ensembles to improve adaptability to out-of-distribution datasets. Additionally, as the task was performed as a simple classification at the patch-level, incorporating advanced methods such as CLAM (Lu et al., 2021) or multiple instance learning-based approaches could potentially improve performance. Another limitation lies in the use of simple probability map averaging as the ensemble method, which may not be optimal. For instance, weighting could be applied to the ensemble probability maps, or a Teacher-Student network could be trained to learn from the ensembled results. Moreover, integrating feature selection from foundation models with feature concatenation may enhance representational capacity. Further exploration of alternative ensemble strategies for foundation models is needed to identify more effective approaches.

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

## Appendix A. Computational Cost Comparison of Various Models

|  | Num params | GFLOPs | Size(MB) | Depth |
|---|---|---|---|---|
| ResNet34 | 21.80M | 3.68B | 83.15 | 54 |
| ViT-Base | 86.57M | 16.87B | 330.22 | 38 |
| Ciga et al. | 11.18M | 1.83B | 42.63 | 29 |
| Ctransapth | 27.52M | 4.51B | 104.98 | 56 |
| HIPT | 21.67M | 6.15B | 82.64 | 49 |
| Phikon | 85.80M | 17.58B | 327.29 | 49 |
| Virchow | 631.23M | 162.07B | 2.41K | 129 |
| Pathoduet | 85.80M | 16.95B | 327.3 | 49 |
| PathDINO | 9.56M | 13.43B | 36.47 | 21 |
| UNI | 303.35M | 59.70B | 1.16K | 97 |
| CONCH | 90.39M | 69.07B | 344.82 | 51 |
| Hibou | 85.74M | 23.56B | 327.07 | 49 |
| Prov-GigaPath | 1.13B | 223.45B | 4.33K | 161 |
| BEPH | 85.76M | 17.58B | 327.15 | 49 |

