# OpenReview forum: "Foundation Model Ensemble for Out-of-Distribution Generalization: Predicting Lymph Node Metastasis in Early Gastric Cancer Using Whole-Slide Imaging"
_MIDL.io/2025/Conference — MIDL 2025 Poster_

### Official Review · Reviewer_EWMk · 2025-02-20

**Confidence:** 4
**Preliminary Rating:** 4
**Recommendation:** Poster
**Final Rating:** 4

**Summary:**

This paper investigates the use of foundation model ensembles to improve out-of-distribution generalization for predicting lymph node metastasis (LNM) in early gastric cancer using whole-slide imaging (WSI). The authors leverage 13 pre-trained models—including one ImageNet-based model and 12 specialized computational pathology models—and evaluate three ensemble strategies: majority voting, averaging probability maps, and feature concatenation. Their experiments, conducted on three distinct datasets (internal surgical, external surgical, and endoscopic resection cases), show that averaging probability maps from the top-3 models not only enhances the AUC scores but also reduces model uncertainty, as quantified by the negative log-likelihood. The study demonstrates that combining diverse representations from different foundation models can mitigate the effects of imaging variability, thereby offering a robust framework for computational pathology applications.

**Strengths:**

The paper presents a well-motivated approach to address the significant challenge of out-of-distribution generalization in WSI analysis. Its strengths lie in the comprehensive evaluation of diverse pre-trained models and the innovative ensemble strategies explored. The methodology is described in detail—from ROI extraction and patch-level processing to various ensembling techniques—which enhances reproducibility. Moreover, the study employs rigorous evaluation metrics (AUC and NLL) and uses multiple heterogeneous datasets, thereby providing robust evidence of the ensemble’s effectiveness. The clear demonstration that averaging probability maps can both improve prediction performance and reduce uncertainty highlights the potential clinical value of the approach.

**Weaknesses:**

1. The reliance on using foundation models solely as fixed feature extractors may limit their adaptability, and the underperformance of the feature concatenation method suggests that more sophisticated integration techniques could be beneficial.
2. The experimental evaluation would be strengthened by additional ablation studies to assess the impact of fine-tuning versus freezing the foundation models.
3. The discussion on computational costs and scalability is somewhat limited, which could be critical for practical clinical deployment. A deeper comparison with other state-of-the-art multi-instance learning approaches would also help contextualize the novelty and practical advantages of the proposed ensemble methods.

**Detailed Comments:**

The paper is well-organized and the experimental design is comprehensive, with a clear explanation of the preprocessing, model architectures, and ensemble strategies. The inclusion of uncertainty quantification through NLL adds significant value by demonstrating improved calibration of predictions. However, more detailed ablation studies—especially regarding the impact of fine-tuning the foundation models—would further clarify the strengths and limitations of using fixed feature extractors. Additional discussion on computational efficiency and potential deployment challenges would also be beneficial. The figures are informative, though some annotations or comparative visual aids could help in better illustrating the performance differences among the methods.

**Justification Of The Final Rating:**

The authors’ rebuttal has addressed most of my questions. The additional analysis, e.g., the updated computational cost table and dataset-specific model selection, reinforces that the ensemble method, particularly the averaging of probability maps, effectively improves both AUC and uncertainty metrics. Although there remain opportunities for further exploration, the enhancements and clarifications provided support the overall contributions of the work, justifying a final rating of weak accept.

**Justification Of The Preliminary Rating:**

The paper presents a novel and methodologically sound approach to enhancing out-of-distribution generalization in computational pathology through ensemble learning. The experimental results, which demonstrate improved AUC and reduced uncertainty via probability map averaging, are promising and indicate significant potential for clinical application. However, limitations such as the exclusive use of fixed feature extractors and the limited exploration of alternative integration strategies (e.g., the underperforming feature concatenation method) prevent a higher rating. Addressing these issues with additional ablation studies and a more detailed discussion on scalability and computational efficiency would further strengthen the paper’s contributions and overall impact.

**Questions To Address In The Rebuttal:**

1. Could the authors elaborate on how the choice of ensemble method might scale when integrating an even larger number of foundation models?
2. How would the performance and computational cost change if the foundation models were fine-tuned rather than used solely as feature extractors?
3. Can the authors provide more insights into the selection criteria for the top-performing models used in the ensemble, and how sensitive is the ensemble performance to these choices?

---

> ### Author Response · Authors · 2025-03-08
> **Reply to Reviewer EWMk**
>
> We sincerely appreciate the reviewer EWMk’s careful review of our submission and insightful comments.
> With this rebuttal, we aim to clarify our work and have incorporated the reviewers' feedback.
> The modifications are highlighted in green in the revised submission.
>
> **Point to Adress:**
>
> ---
>
> 1. Could the authors elaborate on how the choice of ensemble method might scale when integrating an even larger number of foundation models?
>
> Thank you for the valuable insight. Different ensemble strategies could be applied. One approach is to introduce weighted averaging when combining probability maps, allowing the ensemble to prioritize more reliable models. Additionally, a Teacher-Student network could be trained to learn from ensemble-generated results. Feature selection or extraction could also be introduced to manage complexity and reduce redundancy as the number of models scales.
>
> We have included these considerations into the Discussion section.
>
> ---
>
> 2. How would the performance and computational cost change if the foundation models were fine-tuned rather than used solely as feature extractors?
>
> Fine-tuning foundation models could improve performance by adapting their representations to the specific task, but it significantly increases computational cost, especially for large-scale models.  In our study, we used them as feature extractors due to computational constraints, with a classifier consisting of only three linear layers being trained.
>
> For additional insights into computational cost, we have updated the Appendix to include the computational cost of all foundation models in the following table.
>
>
> | Model          | Num Params | GFLOPs  | Size (MB) | Depth |
> |---------------|------------|---------|-----------|-------|
> | ResNet34      | 21.80M     | 3.68B   | 83.15     | 54    |
> | ViT-Base      | 86.57M     | 16.87B  | 330.22    | 38    |
> | Ciga et al.   | 11.18M     | 1.83B   | 42.63     | 29    |
> | CTransPath    | 27.52M     | 4.51B   | 104.98    | 56    |
> | HIPT         | 21.67M     | 6.15B   | 82.64     | 49    |
> | Phikon       | 85.80M     | 17.58B  | 327.29    | 49    |
> | Virchow      | 631.23M    | 162.07B | 2.41K     | 129   |
> | PathoDuet    | 85.80M     | 16.95B  | 327.3     | 49    |
> | PathDINO     | 9.56M      | 13.43B  | 36.47     | 21    |
> | UNI          | 303.35M    | 59.70B  | 1.16K     | 97    |
> | CONCH        | 90.39M    | 69.07B    | 344.82    | 51    |
> | Hibou        | 85.74M     | 23.56B  | 327.07    | 49    |
> | Prov-GigaPath| 1.13B      | 223.45B | 4.33K     | 161   |
> | BEPH         | 85.76M     | 17.58B  | 327.15    | 49    |
>
>
> ---
>
> 3. Can the authors provide more insights into the selection criteria for the top-performing models used in the ensemble, and how sensitive is the ensemble performance to these choices?
>
> In this study, we selected the top-performing models for ensembling based on their performance on the internal validation dataset.
> However, when selecting the top models separately for each dataset, the results changed:
>
> - External dataset: BEPH (0.796), CTransPath (0.778), Hibou (0.779) → AUC 0.824
> - ESD dataset: CTransPath (0.702), CONCH (0.688), Toronto (0.663) → AUC 0.737
>
> These results show improved performance compared to selecting models based on the internal dataset (0.806 for external, 0.714 for ESD), indicating that ensemble performance is sensitive to model selection and that dataset-specific model selection can further enhance performance.
>
> In our study, we used AUC to select the top models (top-3, top-5), but an alternative approach could consider each model’s Brier Score and Negative Log Likelihood for model selection in each dataset.
>
> *Brier Score was additionally computed in this version to assess model calibration.
>
>
> ---
>
> Thank you for your valuable feedback. We hope the above addresses your points. Please let us know if there is anything else we can clarify.

---

### Official Review · Reviewer_MEdL · 2025-02-21

**Confidence:** 5
**Preliminary Rating:** 3
**Final Rating:** 4

**Summary:**

The authors study the problem of using foundation models to predict lymph node metastasis from whole-slide imaging. They have selected 12 recent pathology foundation models and demonstrate the performance of each on 3 datasets. They further study the effectiveness of ensemble the predictions from different foundation models.

**Strengths:**

The study is practical and can serve as a guidance for future work for this task. The authors have conducted a thorough study by including 12 recent foundation models, tested on 3 datasets. Analysis was conducted on accuracy, uncertainty, and consistency.

**Weaknesses:**

1. The clarity of the pipeline can be improved. It looks like the classification involves two steps (1) extract ROI from WSI; (2) Predict LNM based on ROI, and for the 1st step, the authors use the foundation model by Ciga et al. Does this introduce a bias towards Ciga et al.? Specifically, I'm curious to see if Ciga et al. will remain the top performing model is step 1 is based on other foundation models.
2. The authors compare to ensemble of the top-3 and top-5. Can authors also compare to ensemble of the top-1 foundation model and/or baselines  (by repeating the experiment n times and integrate the predictions with voting or averaging probability map)? This can show if having several variations of the same model can be better than integrating the top-k foundation models.
3. If time permits, can the authors study optimizing both the foundation model and classification layer, instead of only using foundation model as feature extractors?

**Detailed Comments:**

Rather minor, but I prefer Table 3 (with extra columns showing difference between current method and baselines) over Figure 2 when showing the performance of each method.

**Justification Of The Final Rating:**

The authors have addressed my second concern by conducting the required experiment and showed the benefit of integrating outputs from different models over from the same one. They have also alleviated my first concern by showing that the outputs from Ciga et al. are verified by a radiologist. Overall I think the systematic study of pathology foundation models can be useful to the community.

**Justification Of The Preliminary Rating:**

The paper studies an interesting problem of utilizing different foundation models for predicting LNM from WSI. There are some concerns on the experimental design and competing strategies. Overall I'm leaning towards acceptance if the concerns are addressed.

**Questions To Address In The Rebuttal:**

See the weakness section.

---

> ### Author Response · Authors · 2025-03-08
> **Reply to Reviewer MEdL**
>
> We sincerely appreciate the reviewer MEdL’s careful review of our submission and insightful comments.
> With this rebuttal, we aim to clarify our work and have incorporated the reviewers' feedback.
> The modifications are highlighted in green in the revised submission.
>
> **Point to Adress:**
>
> ---
>
> 1. The clarity of the pipeline can be improved. It looks like the classification involves two steps (1) extract ROI from WSI; (2) Predict LNM based on ROI, and for the 1st step, the authors use the foundation model by Ciga et al. Does this introduce a bias towards Ciga et al.? Specifically, I'm curious to see if Ciga et al. will remain the top performing model is step 1 is based on other foundation models.
>
> To improve clarity, we have substantially revised the paper based on Reviewer Xa6N’s feedback.
>
> Additionally, the reviewer raised concerns about potential bias introduced by using the foundation model by Ciga et al. for cancer region extraction. However, a pathologist reviewed the cancer ROIs for the entire dataset and confirmed their appropriateness.
>
> We have incorporated these considerations to Section 2.3 (Cancer Region Extraction) for clarity.
>
> ---
>
> 2. The authors compare to ensemble of the top-3 and top-5. Can authors also compare to ensemble of the top-1 foundation model and/or baselines (by repeating the experiment n times and integrate the predictions with voting or averaging probability map)? This can show if having several variations of the same model can be better than integrating the top-k foundation models.
>
> Thank you for the valuable insight. We have added the results of voting and averaging probability maps for the top-1 foundation model and baselines in the Results of Foundation Model Ensembles section. The results show that ensembling multiple foundation models outperforms ensembling a single model.
>
> The table below summarizes the AUC performance, comparing single-model ensembles with a top-3 ensemble:
>
> | Method | Internal | External | ESD |
> |----------------------------------------------|------------|------------|------------|
> | Scratch ResNet34 | 0.839 | 0.748 | 0.538 |
> | Scratch ResNet34 (Voting) | 0.809 | 0.747 | 0.559 |
> | Scratch ResNet34 (Averaging Probability Maps) | 0.811 | 0.747 | 0.556 |
> | Ciga et al. | 0.867 | 0.742 | 0.663 |
> | Ciga et al. (Voting) | 0.863 | 0.741 | 0.639 |
> | Ciga et al. (Averaging Probability Maps) | 0.864 | 0.743 | 0.638 |
> | Top-3 Ensemble (Averaging Probability Maps) | 0.879 | 0.806 | 0.714 |
>
>
>
> ---
>
> 3. If time permits, can the authors study optimizing both the foundation model and classification layer, instead of only using foundation model as feature extractors?
>
> Due to time and resource constraints for this rebuttal, we were unable to conduct additional experiments optimizing both the foundation model and classification layer. Nevertheless, we recognize the potential benefits of this approach and consider it a valuable direction for future research.
>
> ---
> - (Detailed Comments) Rather minor, but I prefer Table 3 (with extra columns showing difference between current method and baselines) over Figure 2 when showing the performance of each method.
>
> Thank you for your suggestions. We agree with the reviewer and have replaced Figure 2 with Table 3 to improve clarity.
>
> ---
>
> Thank you for your valuable feedback. We hope the above addresses your points. Please let us know if there is anything else we can clarify.

---

> > ### Comment · Reviewer_MEdL · 2025-03-14
> >
> > Thank the authors for the detailed response. My concern for 1 is alleviated but still remains; in particular I think the outputs, even confirmed by radiologists, can contain some nuances that are beneficial to Ciga et al. An ideal solution would be to use the same model for both extracting ROI and predicting LNM.
> >
> > However, I understand that this approach can be impractical, and since the authors have addressed my other concerns, I'm happy to raise my score to weak accept.

---

### Official Review · Reviewer_Xa6N · 2025-02-22

**Confidence:** 5
**Preliminary Rating:** 1

**Summary:**

This paper investigates the use of foundation model ensembles for predicting lymph node metastasis (LNM) in early gastric cancer from whole-slide images (WSIs).  The proposed approach involves systematic evaluation of how combining multiple pre-trained foundation models can improve generalization performance compared to individual models, particularly when dealing with out-of-distribution data.
The authors evaluated 13 different pre-trained models (12 foundation models and one ImageNet pre-trained model) across three distinct datasets: internal surgical cases, external surgical cases, and endoscopic submucosal dissection cases. Their ensemble approach, particularly the averaging of probability maps from the top-3 performing models, demonstrated consistent improvements in performance across all datasets.

**Strengths:**

- Inclusion of the most recent and popular foundation models in for performance assessment
- Careful consideration of different ensemble strategies, including majority voting, probability map averaging, and feature concatenation.
- Including both internal and external validation datasets for model evaluation

**Weaknesses:**

The primary concerns center on methodological clarity and justification, particularly regarding the definition of "out-of-distribution" data and the characterization of model methodologies as distinct when many use similar approaches. The experimental design choices, especially those related to image resolution selection and patch processing, require clearer rationale. Additionally, the paper would benefit from more comprehensive evaluation metrics, including computational complexity analysis and proper quantification of probability calibration claims. The presentation of results could be enhanced through better and more legible visualizations.

**Detailed Comments:**

1. The authors did not define what out-of-distribution means for this paper. The authors are advised to refer Farquhar and Gal's paper on "What Out of Distribution is and is not?" (https://openreview.net/pdf?id=XCS_zBHQA2i) for defining the context of the task described in this paper.

2. Most of the foundation models use DINOv2 as the learning algorithm and vision transformer as the model, so what do the authors mean by "distinct methodologies" in the context of pathology foundation models?

3. Please create a table to show the data distribution and characteristics across training, internal validation, and testing sets. This will improve readability of the paper.

4. Regarding Section 2.3
4a. Why did the authors rely on ROI based approach (Section 2.3) for training and evaluating the performance of the foundation models for LNM prediction?
4b. When the input WSIs were scanned at 40x optical resolution, why did the authors use 20x, 10x, and 5x resolution - why not use 40x patches as well?
4c. Were there three resolution specific models trained for LNM prediction for each foundation model used in this paper?

5. Regarding Section 2.4
5a. For LNM prediction why patches from 10x magnification were extracted? Why to discard information available at 40x or 20x or 5x?
5b. Why not extract 224x224 sized patches to begin with instead of extracting 512x512 patches initially followed by random cropping?
5c. What happened to the models trained on 20x and 5x resolution as mentioned in Section 2.3? Why only the 10x model was used?

6. Section 2.5 is confusing - On one hand the authors state that they trained three non-linear layers for classification while on the other they mentioned that ResNet34 and ViT-Base models were trained from scratch. The reviewer is not sure what was the actual classification model here?

7. Why top 100 patches from the heatmap was used for the final classification? why not top 10, 20, or 500 or any other number? This choice seems arbitrary.

8. Please avoid writing inline equations (exampe, the negative log likelihood equation) as these are difficult to cross-refer.

8. The clarity about the horizontal lines for Resnet34 and ViT-base models is missing. Not sure why someone should use foundation models when Resnet-34 alone is performing pretty good here.

9. Figure 3a is too small to draw any meaningful conclusion from it. The authors are advised to show larger heatmaps for clarity.

10. In the caption of Figure 3 the authors wrote "..ensembling leads to more calibrated probability distribution.." The reviewer is however not sure how the authors measured the calibration of the ensemble model. How do the authors compute the underlying true probability distribution for patch-level LNM classification to assess model calibration here?

11. In Table 2, the benchmarks should aslo include the independent best foundation model results for each set.

12. Computational Complexity analysis is completely missing in the paper. The authors propose to use an ensemble of foundation models for producing better results but they should also include a section about the computational cost at which those better results will be generated and whether the performance improvement is worth the computational cost.

**Justification Of The Preliminary Rating:**

Based on the reviewer comments this paper needs a major revision which are beyond just organizational or cosmetic changes. The paper requires significant clarity on the proposed methodology and several details about the choice of hyper-parameters used in the paper are missing. Thus, the reviewer thinks that this paper should be rejected in its current form.

**Questions To Address In The Rebuttal:**

1. The authors did not define what out-of-distribution means for this paper. The authors are advised to refer Farquhar and Gal's paper on "What Out of Distribution is and is not?" (https://openreview.net/pdf?id=XCS_zBHQA2i) for defining the context of the task described in this paper.

2. Most of the foundation models use DINOv2 as the learning algorithm and vision transformer as the model, so what do the authors mean by "distinct methodologies" in the context of pathology foundation models?

3. Please create a table to show the data distribution and characteristics across training, internal validation, and testing sets. This will improve readability of the paper.

4. Regarding Section 2.3
4a. Why did the authors rely on ROI based approach (Section 2.3) for training and evaluating the performance of the foundation models for LNM prediction?
4b. When the input WSIs were scanned at 40x optical resolution, why did the authors use 20x, 10x, and 5x resolution - why not use 40x patches as well?
4c. Were there three resolution specific models trained for LNM prediction for each foundation model used in this paper?

5. Regarding Section 2.4
5a. For LNM prediction why patches from 10x magnification were extracted? Why to discard information available at 40x or 20x or 5x?
5b. Why not extract 224x224 sized patches to begin with instead of extracting 512x512 patches initially followed by random cropping?
5c. What happened to the models trained on 20x and 5x resolution as mentioned in Section 2.3? Why only the 10x model was used?

6. Section 2.5 is confusing - On one hand the authors state that they trained three non-linear layers for classification while on the other they mentioned that ResNet34 and ViT-Base models were trained from scratch. The reviewer is not sure what was the actual classification model here?

7. Why top 100 patches from the heatmap was used for the final classification? why not top 10, 20, or 500 or any other number? This choice seems arbitrary.

8. Please avoid writing inline equations (exampe, the negative log likelihood equation) as these are difficult to cross-refer.

8. The clarity about the horizontal lines for Resnet34 and ViT-base models is missing. Not sure why someone should use foundation models when Resnet-34 alone is performing pretty good here.

9. Figure 3a is too small to draw any meaningful conclusion from it. The authors are advised to show larger heatmaps for clarity.

10. In the caption of Figure 3 the authors wrote "..ensembling leads to more calibrated probability distribution.." The reviewer is however not sure how the authors measured the calibration of the ensemble model. How do the authors compute the underlying true probability distribution for patch-level LNM classification to assess model calibration here?

11. In Table 2, the benchmarks should aslo include the independent best foundation model results for each set.

---

> ### Author Response · Authors · 2025-03-08
> **Reply to Reviewer Xa6N**
>
> We sincerely appreciate reviewer Xa6N’s careful review of our submission and insightful comments. This constructive critique has helped us enhance the clarity of the paper. We have addressed all concerns in the rebuttal and revised the manuscript accordingly. The modifications are highlighted in green in the revised submission.
>
> **Point to Adress:**
>
> ---
> 1. The authors did not define what out-of-distribution means for this paper. The authors are advised to refer Farquhar and Gal's paper on "What Out of Distribution is and is not?" (https://openreview.net/pdf?id=XCS_zBHQA2i) for defining the context of the task described in this paper.
>
> Thank you for the valuable advice. We have included the referenced literature and clarified the definition of out-of-distribution in the Introduction section.
>
> ---
> 2. Most of the foundation models use DINOv2 as the learning algorithm and vision transformer as the model, so what do the authors mean by "distinct methodologies" in the context of pathology foundation models?
>
> We sincerely thank the reviewer for this insightful comment. We acknowledge that the original phrasing,
>
> "*Since each model was trained on diverse datasets using distinct methodologies, they learn unique representations...*"
>
> may have been ambiguous. By "distinct methodologies," we intended to highlight variations in architectural adjustments, learning methods, and differences in training procedures across models.
>
> To enhance clarity and avoid confusion, we have revised the sentence as follows:
>
> "*Since each foundation model was trained on datasets collected from different sources under varying settings, the learned representations reflect different characteristics to some extent.*"
>
> ---
> 3. Please create a table to show the data distribution and characteristics across training, internal validation, and testing sets. This will improve readability of the paper.
>
> Thank you for the suggestion. We have added a table to Section 2.1(Datasets) to provide an overview of the datasets.
>
> ---
> 4. Regarding Section 2.3 4a. Why did the authors rely on ROI based approach(Section 2.3) for training and evaluating the performance of the foundation models for LNM prediction? 4b. When the input WSIs were scanned at 40x optical resolution, why did the authors use 20x, 10x, and 5x resolution - why not use 40x patches as well? 4c. Were there three resolution specific models trained for LNM prediction for each foundation model used in this paper?
>
> We sincerely apologize for any confusion caused. To improve clarity, we have rephrased the Method section accordingly.
>
> - 4a. Lymph node metastasis (LNM) is closely associated with tumor characteristics. We have cited relevant literature.
> - 4b. Using 40× resolution would have significantly increased processing time. Instead, we combined results from 20×, 10×, and 5× resolutions. A pathologist reviewed the extracted cancer regions at these magnifications and confirmed their appropriateness.
> - 4c. The three resolution-specific models were used only for cancer region extraction, not for LNM prediction. For LNM prediction, we used a single resolution. We have revised this section for clarity.
>
> ---
>
> 5. Regarding Section 2.4 5a. For LNM prediction why patches from 10x magnification were extracted? Why to discard information available at 40x or 20x or 5x? 5b. Why not extract 224x224 sized patches to begin with instead of extracting 512x512 patches initially followed by random cropping? 5c. What happened to the models trained on 20x and 5x resolution as mentioned in Section 2.3? Why only the 10x model was used?
>
> - 5a. While 20× and 40× could provide more details, they would have significantly increased computational costs. We extracted 60,097 patches (512×512 at 10× magnification) o train a model for LNM prediction. At 10× resolution, training took over a day per model due to the large number of patches, and higher magnifications would have further multiplied this time—approximately fourfold (20×) and sixteenfold (40×).
> - 5b. The foundation models used in this paper have varying input sizes, including 224, 256, 448, and 512. To address this, we extracted 512×512 patches and applied random cropping to match each model’s required input size. We have clarified this process in the revised manuscript.
>
> The original sentence stated:
>
> "*...with a size of 512×512 pixels and an overlap ratio of 0.5 between tiles. Subsequently, the tiles were randomly cropped into patches to match the input size required by each foundation model.*"
>
> To improve clarity, we have revised it to:
>
> "*Since each foundation model requires a different input image size based on its trained settings, we tiled the cancer regions into 512×512 pixels with an overlap ratio of 0.5 between adjacent tiles.  Subsequently, the tiles were randomly cropped...*"
>
> - 5c. For LNM prediction, only the 10× resolution model was used. The 20× and 5× models were used only for cancer region extraction. We have revised the Method section for clarity.

---

> ### Author Response · Authors · 2025-03-08
> **Reply to Reviewer Xa6N**
>
> Due to character limitations, we are continuing our response below.
>
> 6. Section 2.5 is confusing - On one hand the authors state that they trained three non-linear layers for classification while on the other they mentioned that ResNet34 and ViT-Base models were trained from scratch. The reviewer is not sure what was the actual classification model here?
>
> We sincerely apologize for any confusion. To improve clarity, we have moved the relevant content from Section 2.5 to Section 2.4 and rephrased it as follows.
>
> The original sentence stated:
>
> "*To predict LNM from WSI, foundation models were used as feature extractors, and classification networks consisting of three non-linear layers was trained at the patch-level. ResNet34 (He et al., 2015) and ViT-Base (Dosovitskiy et al., 2020) served as baseline models, trained from scratch.*"
>
> The revised version:
>
> "*For baseline comparisons, ResNet34 (He et al., 2015) and ViT-Base (Dosovitskiy et al., 2020) were trained from scratch. The foundation models were used as feature extractors, and for each, a patch-level classifier consisting of three non-linear layers with a sigmoid function at the output was trained.*"
>
> ---
>
> 7. Why top 100 patches from the heatmap was used for the final classification? why not top 10, 20, or 500 or any other number? This choice seems arbitrary.
>
> The top-100 patch selection was determined experimentally by testing various criteria, including max, top-10, top-20, top-50, top-100, and percentile-based selections (top 25%, 50%, 75%). We have updated the paper as follows:
>
> "*Slide-level LNM predictions were obtained by averaging the top-100 patches, selected experimentally.*"
>
> ---
>
> 8. Please avoid writing inline equations (exampe, the negative log likelihood equation) as these are difficult to cross-refer.
>
> Thank you for your suggestion. We have revised the manuscript to avoid inline equations.
>
> ---
>
> 9. The clarity about the horizontal lines for Resnet34 and ViT-base models is missing. Not sure why someone should use foundation models when Resnet-34 alone is performing pretty good here.
>
> Thank you for the feedback. To improve clarity, we have replaced the figure with the table below.
>
> | **Model Name**     | **Internal Dataset** | **External Dataset** | **ESD Dataset** |
> |--------------------|----------------------|----------------------|-----------------|
> | Scratch ResNet34   | 0.839                | 0.748                | 0.538           |
> | Scratch ViT-base   | 0.827                | 0.786                | 0.471           |
> | ImageNet (ResNet34) | 0.794                | 0.760                | 0.641           |
> | Ciga et al.        | **0.867**                | 0.742                | 0.663           |
> | CTransPath         | 0.859                | 0.778                | **0.702**           |
> | HIPT               | 0.802                | 0.701                | 0.600           |
> | Phikon             | 0.841                | 0.763                | 0.628           |
> | Virchow            | 0.858                | 0.754                | 0.655           |
> | Pathoduet          | 0.740                | 0.734                | 0.510           |
> | PathDINO           | 0.847                | 0.700                | 0.534           |
> | UNI                | 0.853                | 0.718                | 0.631           |
> | CONCH              | 0.816                | 0.700                | 0.688           |
> | Hibou              | 0.843                | 0.779                | 0.597           |
> | GigaPath           | 0.835                | 0.646                | 0.621           |
> | BEPH               | 0.827                | **0.796**                | 0.604           |
>
> ---
>
> 10. Figure 3a is too small to draw any meaningful conclusion from it. The authors are advised to show larger heatmaps for clarity.
>
> Thank you for the suggestion. We have included an enlarged heatmap in the revision.
>
> ---
>
> 11. In the caption of Figure 3 the authors wrote "..ensembling leads to more calibrated probability distribution.." The reviewer is however not sure how the authors measured the calibration of the ensemble model. How do the authors compute the underlying true probability distribution for patch-level LNM classification to assess model calibration here?
>
> Thank you for the valuable insight. To assess calibration, we have included the Brier score and updated the Method and Results sections accordingly.
>
> ---
>
> 12. In Table 2, the benchmarks should aslo include the independent best foundation model results for each set.
>
> We have updated Table 2 to include the best-performing individual foundation model results for each dataset.
>
> ---
>
> We greatly appreciate the reviewer's valuable feedback and detailed suggestions. We hope the above addresses your points. Please let us know if there is anything else we can clarify.

---

### Author Rebuttal · Authors · 2025-03-08

**Rebuttal:**

We sincerely appreciate the reviewers' insightful and constructive feedback, which has been invaluable in enhancing the clarity and quality of our manuscript. We have carefully considered each comment and have made substantial revisions to address the concerns raised. The modifications are highlighted in green in the revised submission.

Below is a summary of our responses to the key points:

- We received feedback from the reviewers regarding the clarity of our paper. In response, we have restructured and revised the manuscript to improve clarity.

- We have replaced Figure 2, which showed the performance of each model, with table for better clarity.

- We have added the computational cost of each model to the appendix.

- We have added the ensemble results of a single model for comparison.

- To assess model calibration, we have added an evaluation using the Brier score.

Additionally, we would like to inform the reviewer that we discovered an error in our paper after submission. The current evaluation metric is AUC, but majority voting produces binary results, which are not suitable for AUC calculation. Therefore, we have replaced the foundation model ensemble results, changing from majority voting to soft voting.

Once again, we sincerely appreciate the reviewers' detailed suggestions.

**Supporting Material:**

/attachment/a8bf56b12fb8841fd519dcb2ec9517ae3e6f8dd9.pdf

---

### Meta-Review · Area_Chair_9zRp · 2025-03-23

**Recommendation:** Accept (Poster)
**Confidence:** 3

**Metareview:**

The paper receives mixed reviews (two positive and one strong negative).  Both positive reviewers agree that authors have conducted a thorough study by including 12 recent foundation models, and the study is practical and may serve as a guidance for future work for this task.  I also agree with reviewer Xa6N that there are issues with the methodological clarity/justification and the experiment clarity.  However, considering this paper is in application track, I lead to accept it.